# Physical measures of physical functioning as prognostic factors to predict outcomes in low back pain: Protocol for a systematic review

**Rameeza Rashed**[ID]*, **Katie Kowalski**[ID], **David Walton**, **Afieh Niazigharemakhe**, **Alison Rushton**[ID]

School of Physiotherapy, Western University, London, Ontario, Canada

* rrashed2@uwo.ca

**Data Availability Statement:** Data will be published in the paper.

## Abstract

### Background

Low back pain (LBP) is a highly prevalent condition that substantially impairs individuals' physical functioning. This highlights the need for effective management strategies to improve patient outcomes. It is, therefore, crucial to have knowledge of physical functioning prognostic factors that can predict outcomes to facilitate the development of targeted treatment plans aiming to achieve better patient outcomes. There is no synthesis of evidence for physical functioning measures as prognostic factors in the LBP population. The objective of this systematic review is to synthesize evidence for physical measures of physical functioning as prognostic factors to predict outcomes in LBP.

### Methods

The protocol is registered in the International Prospective Register of Systematic Reviews and reported in line with the Preferred Reporting Items for Systematic Review and Meta-Analysis Protocols (PRISMA-P). Prospective longitudinal observational studies investigating potential physical prognostic factors in LBP and/or low back-related leg pain population will be included, with no restriction on outcome. Searches will be performed in MEDLINE, EMBASE, Scopus, CINAHL databases, grey literature search using Open Grey System and ProQuest Dissertations and Theses, hand-searching journals, and reference lists of included studies. Two independent reviewers will evaluate the eligibility of studies, extract data, assess risk of bias and quality of evidence. Risk of bias will be assessed using the Quality in Prognostic Studies (QUIPS) tool. Adequacy of clinical, methodological, and statistical homogeneity among included studies will decide quantitative (meta-analysis) or qualitative analysis (narrative synthesis) focused on prognostic factors and strength of association with outcomes. Quality of cumulative evidence will be evaluated using a modified Grading of Recommendations Assessment, Development, and Evaluation (GRADE).

**Funding:** The authors received no specific funding for this work.

**Competing interests:** The authors have declared that no competing interests exist.

## Discussion

Information about prognostic factors can be used to predict outcomes in LBP. Accurate outcome prediction is essential for identifying high-risk patients that allows targeted allocation of healthcare resources, ultimately reducing the healthcare burden.

## Registration

PROSPERO, CRD42023406796.

## Introduction

Low back pain (LBP) has been a prevalent and persistent global health issue for the past three decades [1]. It continues to rank among the highest causes of Years Lived with Disability, contributing to a decline in functionality throughout the lifespan and imposing a substantial economic burden [1, 2]. Prognostic research enables early identification of people at risk for poor outcomes, facilitating informed decision-making for timely and targeted treatment interventions that aim to improve outcomes and reduce disability burden [3]. A prognostic factor refers to a measurement or characteristic that can predict subsequent health outcomes in the context of a particular health condition, aiding in understanding and anticipating the natural course of the condition [4]. It is, therefore, crucial to have knowledge of prognostic factors that can predict outcomes in LBP for effective management.

The Core Outcome Measures in Effectiveness Trials Initiative (COMET) defines physical functioning as the impact of a disease or condition on physical activities of daily living, such as walking, self-care, performance status and disability index [5]. Physical functioning is a multifaceted construct that encompasses various interconnected dimensions. These dimensions include body structure and function, performance of physical activities, social and role functioning [6]. These dimensions interact and influence each other, forming a comprehensive picture of a person's physical functioning. Limitations in one dimension may impact another, resulting in an increased risk of disability and decreased quality of life [7, 8].

Assessment of physical functioning can be categorized based on recommendations from the Initiative on Methods, Measurement, and Pain Assessment in Clinical Trials (IMMPACT), evaluating structure or function of a specific body part or system (e.g., muscle endurance, range of motion) [9], evaluating performance on a defined task in a standardized environment (e.g., 6 min walk test) [10], and evaluating activity in an individual's natural environment (e.g., accelerometery) [9, 10]. Measures of physical functioning offer valuable insight into the broader impact of adverse health conditions on a person's quality of life, which is a significant concern for the affected individuals [7, 11]. The core outcome set to measure the effectiveness of interventions for LBP also recommends physical functioning as an outcome domain that should be reported in all clinical trials [7]. Physical functioning has been assessed using both patient-reported outcome measures (PROMs) [9] and physical outcome measures [12] such as performance-based outcome measures [13] or clinician-administered outcome measures [14] which offer complementary yet distinct information [15]. PROMs offer insights into patients' perceived functional well-being [16], their usefulness can be limited by susceptibility to biases arising from patients' beliefs about their performance or abilities [14], as well as contextual and psychosocial factors such as pain and emotional state [17]. Physical outcome measures can provide more detailed information about specific aspects of physical functioning, such as

walking speed and muscle strength [18]. Owing to the emergence of physical functioning assessment as a key component of multidimensional evaluation during the last two decades [19], several physical measures of physical functioning as prognostic factors have been identified, and are increasingly being adopted in clinical and research settings [20].

Previous systematic reviews examining prognostic factors in LBP have considered a range of factors, encompassing personal, modifiable, unmodifiable, physical (such as X-ray findings), psychological, and work-related factors [15, 16]. However, a notable gap in the existing literature is the investigation of physical measures of physical functioning as prognostic factors for LBP outcomes. Two systematic reviews to date have included physical measures of physical functioning as prognostic factors in LBP. Hartvigsen et al focused on clinical tests as prognostic factors and reported inconsistent evidence [21]. However, this review was limited to low-tech clinical tests only. The level of confidence in the results of this review was not high due to lack of reporting clarity and methodological shortcomings in several included papers. Consequently, the overall assessment score indicated low-quality based on AMSTAR-2 criteria. Verkerk et al. explored several prognostic factors; however, they inadequately addressed and did not comprehensively cover physical measures of physical functioning. This review focused solely on muscle endurance, strength, and aerobic capacity within the category of physical functioning [22]. The review possesses methodological shortcomings (lacking sufficient details on selection criteria, exhibiting low methodological quality of included studies, and lacking to report on the strength of associations between prognostic factors and outcomes) with a low-quality assessment score (AMSTAR-2 criteria).

This limited investigation of physical measures of physical functioning represents a significant gap in understanding their role as prognostic factors in LBP. Therefore, further research is warranted to comprehensively evaluate physical measures of physical functioning predicting outcomes in LBP. This is of particular importance considering the growing field of prognostic research [23] and increasing significance of using physical measures for physical functioning [24].

## Objective

To synthesize the evidence for physical measures of physical functioning as prognostic factors to predict outcomes in the LBP population.

## Material and methods

### Design

This protocol for a systematic review is designed using the Preferred Reporting Items for Systematic Review and Meta-Analysis Protocols (PRISMA-P) 2015 statement [25] and Cochrane handbook [26]. The protocol is registered in PROSPERO (Registration No. CRD42023406796) on 12 April 2023 and any amendments will be reported.

### Eligibility criteria

**Inclusion criteria.** *Population*. Participants aged 18 years and above with LBP and / or low back-related leg pain.

*Potential physical prognostic factors*. All physical measures of physical functioning that have been investigated as predictors of outcomes, and are practically feasible in terms of time, space, training, safety and cost to perform in hospital or community- based clinic (e.g., Timed up and go, 6 min walk test, accelerometery) will be included. Physical measures of physical functioning will be categorized as the following:

1. Impairment based physical measures, evaluating structure or function of a specific body part or system (e.g., range of motion) [9].

2. Performance-based measures, evaluating performance on a defined task in standardized environment (e.g., 6 min walk test) [10].

3. Physical measures of activity in natural environment/real-world, evaluating activity in natural environment (e.g., accelerometery) [10].

*Outcomes*. Any outcome predicted by physical measures of physical functioning (intentional broad definition of outcome following scoping search that identified a limited number of studies).

*Timing and setting*. Any time point for outcome assessment.

*Study design*. Prospective longitudinal observational studies, as they are considered the gold standard for prognostic research enabling optimal measurement of outcomes [27].

**Exclusion criteria.** Any study that includes LBP related to malignancy, fracture, infection, cauda equina, rheumatoid arthritis, and ankylosing spondylitis will be excluded. Studies published in any language except English will be excluded. Physical measures such as imaging, electrophysiological measures (EMG), and motion capture gait analysis utilizing force plates and three-dimensional video analysis will not be included.

## Information sources

A comprehensive search will be performed in electronic databases from inception to June 30, 2023 on MEDLINE, EMBASE, CINAHL, and Scopus. The grey literature will be searched using Open Grey System and ProQuest Dissertations and Theses. Hand searches of key journals (Spine, European Spine Journal, The Spine Journal) and screening reference list of included studies will also be performed.

## Search strategy

The search strategy was developed in collaboration with a research librarian around the constructs of LBP, physical measures of physical functioning and prognostic factors. Search terms were informed by the National Institute for Health and Care Excellence guidelines for LBP and sciatica in over 16s [28], previous systematic reviews focused to physical measures of physical functioning [29] and a search filter developed by the Delphi panel for Ovid MEDLINE to identify prognostic factor studies [30]. The search was developed in MEDLINE and will be adapted for use in other databases. The MEDLINE search strategy is provided in S1 File.

## Data management

The citations retrieved from searches will be imported and archived into Covidence, a web-based software platform designed for the purposes of conducting systematic reviews. The software will automatically detect any duplicate records and remove them accordingly. Following the initial screening of titles and abstracts, the full texts of relevant citations will be uploaded and securely stored within the Covidence platform. Eligibility screening, both at the title and abstract stage, as well as at the full-text stage, will be performed using the Covidence platform.

## Study selection process

Two independent reviewers will perform eligibility assessments for the citations retrieved. During the initial screening phase, titles and abstracts will be evaluated based on predefined eligibility criteria. Full-texts of articles will be obtained for citations that meet the criteria

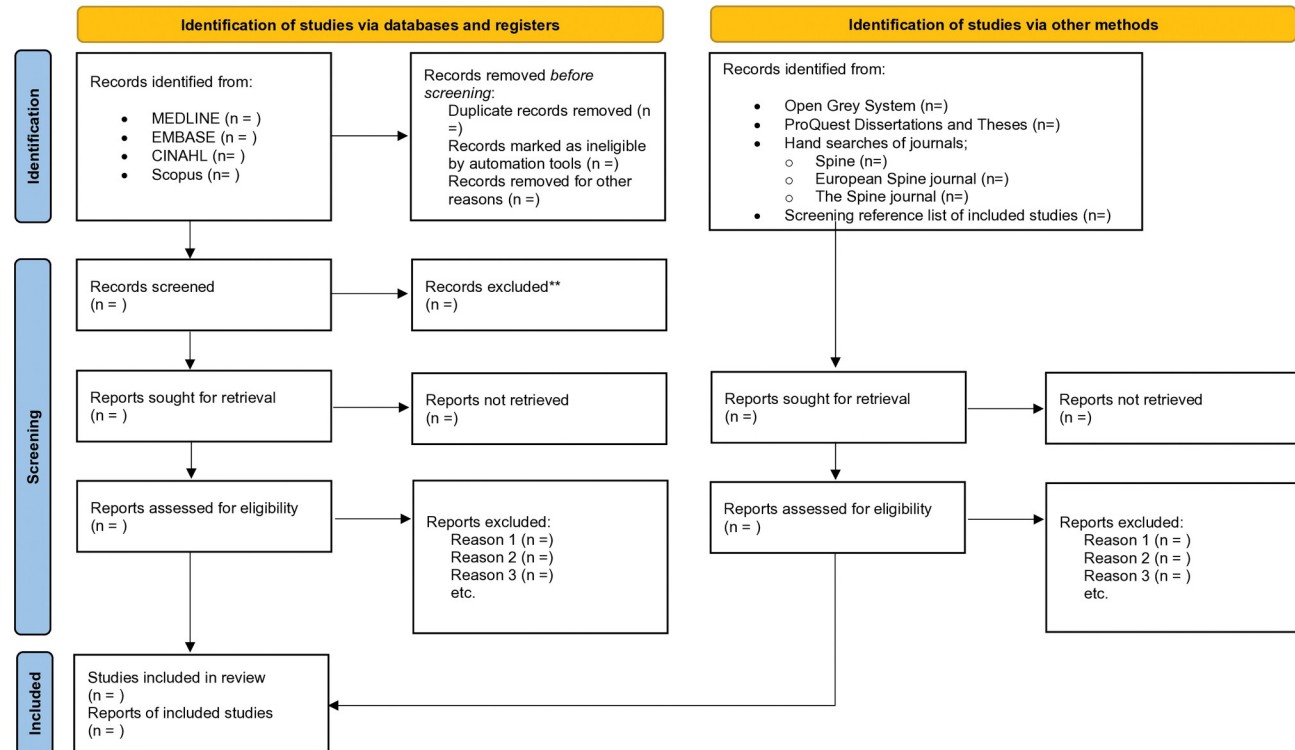

**Fig 1. PRISMA 2020 flow diagram.** This diagram reflects the study selection process.

agreed upon by both reviewers or for articles that lack sufficient information in the title and abstract to determine eligibility. In the subsequent phase, full-texts will be screened independently by the reviewers in duplicate to determine whether the articles meet the eligibility criteria. To ensure consistency in the assessment process, disagreements between the two reviewers will be resolved through discussions. If consensus cannot be reached, a third reviewer will be consulted to resolve the disagreement. The level of agreement between reviewers will be assessed using the Cohen's kappa [31] in Covidence. A PRISMA flowchart will illustrate the selection of included and excluded studies and the reasons for exclusion as shown in the Fig 1 [32].

## Data extraction process

Data will be extracted using a standardized data extraction form by Checklist for Critical Appraisal and Data Extraction for systematic review of prognostic factor studies (CHARMS-PF) [33]. This form for prognostic factors studies has been developed [34], by additions and modifications of the CHARMS checklist for primary studies of prediction models [35]. To ensure the reliability and feasibility of this modified form pilot testing will be performed. Two reviewers will independently extract data from selected studies and ambiguities or challenges in the form will be assessed. Any necessary modifications to the form will be made accordingly. Authors of included studies will be contacted a maximum three times via email in case of any missing data.

**Table 1. Data to be extracted from included studies.**

| Study characteristics | Authors, publication year, study design, country of study, objective of study, source of funding |
|---|---|
| Study participants | Age, gender, LBP history, other health conditions, sample size, missing data |
| Potential physical prognostic factors | Name, type (e.g., performance based), equipment required |
| Outcome measures | Type, equipment required, follow up duration (Time points of assessment of outcome), |
| Results | Analysis method, type of statistical measure, Main findings, other relevant information |

## Data items

Data items will be extracted based on modified CHARMS-PF checklist [34]. Table 1. shows the items to be extracted from the included studies.

## Outcomes and prioritization

As the scoping search identified limited studies and given the broad spectrum of outcome measures used with the LBP population, all outcome measures will be included. The measures can be either PROMs (e.g., ODI) or physical outcome measures (e.g., SLR). Based on the scoping search, a pre-defined outcome of interest and the use of one-time point is not possible owing to variability in the literature. Due to variability in follow-up assessment time points, outcomes will be categorized as short-term (<3 months), medium-term (>3 months, < 12 months), and long-term (>12 months).

## Risk of bias in individual studies

To evaluate the risk of bias (RoB) in the studies included in the review, two independent reviewers will use the QUIPS tool [36], which is recommended by Cochrane for assessing the RoB in prognostic studies. The inter-rater reliability of QUIPS has been demonstrated to be good to acceptable, and previous studies have used QUIPS successfully in prognostic reviews [37, 38]. It consists of multiple prompting items categorized into six domains: study participation, study attrition, prognostic factor measurement, outcome measurement, study confounding, statistical analysis and reporting. Each domain is graded as low, moderate, or high risk of bias. An overall RoB score will be decided for each study based on the scores of all domains. If all six domains are rated as low RoB, or no more than one is rated as moderate ROB, the study will be categorized as low RoB. If one or more domains are rated as high RoB, or $\geq 3$ domains are rated as moderate ROB, the study will be assessed as high RoB. All studies in between will be classified as having moderate ROB [39]. Any difference of opinion between the two reviewers will be discussed. If no consensus can be achieved, an impartial third reviewer will be consulted for their decision. Using Cohen's kappa coefficient, the level of agreement between the two reviewers at each step will be evaluated [40].

## Data synthesis

Data synthesis for each prognostic factor will be performed by pooling data quantitively (meta-analysis) or summarizing results qualitatively (narrative synthesis). Selection of data synthesis method will depend on heterogeneity across included studies. Determination of heterogeneity will be based on clinical, methodological and statistical heterogeneity [41]. Clinical heterogeneity will be assessed by variation in study population characteristics, coexisting

conditions, outcomes, outcome measures, follow-up duration, and prognostic factors. Methodological heterogeneity will be based on RoB in included studies and statistical inconsistency will be determined by effect sizes and direction of effect by Cochrane test (significance level of 0.05) supplemented by the $I^2$ statistic [30]. Absence of heterogeneity will be considered if $I^2 <$ 5% and/or Cochrane Q test is non-significant while $I^2$ value less than 25% will be interpreted as low heterogeneity, between 25% and 50% as moderate and exceeding 50% as high heterogeneity [42].

Meta-analysis will be conducted in case of sufficiently homogeneous studies. As we are anticipating widely different methodologies, outcomes, prognostic factors, and unexplained heterogeneity across studies, a random effect approach will be used. This approach will provide a summary estimate of the average prognostic effect of index factor and the variability in effect across studies. The statistical data reported by the original sources will be extracted and evaluated in relation to established thresholds for statistical or clinical significance [43]. Considering the study design being included in this review, the extracted statistics are expected to include odd ratios, relative risks, hazard ratios, beta coefficients, or likelihood ratios, however any other relevant statistics will be extracted per the source author. In case of high heterogeneity (clinical, methodological and statistical heterogeneity) across studies a narrative synthesis will be conducted [38].

Results across studies will be presented in tabular form to compare different groups, including potential prognostic factors (impairment based, performance based and activity in natural environment), strength of association with different outcomes in relation to follow up time duration (short, medium or long term).

## Meta-bias(es)

If protocols are identified within our searches, reporting bias will be evaluated through consistency of study protocols and published results.

## Confidence in cumulative evidence

GRADE (Grading of Recommendations Assessment, Development, and Evaluation) will be used by two reviewers independently, to rate the quality of evidence (level of confidence) in overall findings, and further elaborating strength of effect across all findings for each prognostic factor per outcome. As advised for prognostic factor research a modified GRADE approach proposed by Huguet et al [44] will be used. The modified GRADE consists of six domains (phase of investigation, study limitations, inconsistency, indirectness, imprecision, publication bias) that determine quality of evidence. Quality can be rated as high, moderate, low, or very low depending on these domains.

All studies included in this review are prospective observational studies, hence the evidence will be of moderate quality as a starting point [38]. The quality of evidence in this review may be downgraded based on several factors. First, methodological limitations in individual studies assessed using the QUIPS tool will result in evidence being rated as having no serious limitation, serious limitation, or very serious limitation (Risk of Bias). Second, unexplained variability in results across studies (inconsistency), and third, evidence that does not directly apply to the population, setting, and outcome of interest (indirectness) will decrease the quality of evidence. Fourth, small sample sizes and wide confidence intervals around estimated effect sizes (imprecision) will downgrade evidence quality. Finally, selective reporting or publication bias will also decrease evidence quality [44]. Two factors that can increase the quality of evidence are a moderate or large effect size and an exposure-response gradient. A moderate effect size would be based on a standardized mean difference of around 0.5, while a large effect size

would be around 0.8 or greater [44]. An exposure-response gradient would be demonstrated if higher levels of the prognostic factor led to a larger effect size. In these cases, the confidence in the quality of evidence will be upgraded. Furthermore, strength of effect determined by magnitude of effect estimates will suggest small, medium, large or inconsistent associations between prognostic factor and outcome across all studies. Summary of evidence will include level of confidence on cumulative evidence (high, moderate, low or very low) in strength of association (small, medium, large or inconsistent) of prognostic factors and outcomes across studies [44].

## Patient and public involvement

Patient involvement has informed this protocol through discussion of the planned systematic review with the spinal pain research Patient Partner Advisory Group at the School of Physical Therapy, Western University. Further discussion will inform interpretation of results from the patients' perceptive and stimulate future research initiatives.

## Discussion

This will be a comprehensive and low RoB systematic review providing a summary of evidence for physical measures of physical functioning as prognostic factors for outcomes in the LBP population. This review will address important gaps in the literature, since there are several physical measures of physical functioning that are used in clinical practice in LBP management [45], but knowledge of their association with outcomes is lacking. Aligned with the importance of physical outcome measures in recently updated physical therapy clinical practice guideline for LBP management [46], this review will identify the prognostic value of using these factors in clinical practice for prediction. Knowledge of prognostic factors will benefit the healthcare practitioners, providing valuable insight for decision-making towards patient care and prioritizing appropriate treatment strategies [47]. Healthcare practitioners can use a risk stratification approach to identify and categorize subgroups of patients with low back pain based on their likelihood of experiencing poor or good outcomes. This approach allows for early intervention, targeted allocation of healthcare resources and support to those at higher risk of poor outcomes, ultimately improving patient outcomes and reducing the burden on the healthcare system. Moreover, providing evidence-based information to patients about the probability of developing various outcomes can positively impact the patients' quality of life by managing their expectations for recovery [48]. Further research of physical prognostic factors of physical functioning may involve other regions of spine (e.g., neck and thoracic region).

## Supporting information

**S1 Checklist. PRISMA-P (Preferred Reporting Items for Systematic review and Meta-Analysis Protocols) 2015 checklist: Recommended items to address in a systematic review protocol\*.**
(DOC)

**S1 File. Search strategy.** This file shows MEDLINE search strategy.
(PDF)

## Acknowledgments

Librarian, Western University.

## Author Contributions

**Conceptualization:** Rameeza Rashed, Katie Kowalski, David Walton, Afieh Niazigharemakhe, Alison Rushton.

**Methodology:** Rameeza Rashed, Katie Kowalski, David Walton, Afieh Niazigharemakhe, Alison Rushton.

**Supervision:** Katie Kowalski, David Walton, Alison Rushton.

**Validation:** Katie Kowalski, David Walton, Alison Rushton.

**Writing – original draft:** Rameeza Rashed.

**Writing – review & editing:** Rameeza Rashed, Katie Kowalski, David Walton, Afieh Niazigharemakhe, Alison Rushton.

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
