## [Editor Report · Decision Letter 0]

11 Aug 2023

PONE-D-23-23995

Physical measures of physical functioning as prognostic factors to predict outcomes in low back pain: protocol for a systematic review

PLOS ONE

Dear Dr. Rashed,

Thank you for submitting your manuscript to PLOS ONE. After careful consideration, we have decided that your manuscript does not meet our criteria for publication and must therefore be rejected.

Specifically:

As there are a number of omissions in the design of this study and written English is not academic standard, the value of this topic under review is questionable.

I am sorry that we cannot be more positive on this occasion, but hope that you appreciate the reasons for this decision.

Kind regards,

Hsiu-Ching Chiu, PhD

Academic Editor

PLOS ONE

Additional Editor Comments:

Please see the attachment

- - - - -

---

## [Author Response · Author response to Decision Letter 0]

15 Sep 2023

Thank you for your review and consideration of our manuscript titled "Physical measures of physical functioning as prognostic factors to predict outcomes in low back pain: protocol for a systematic review" (PONE-D-23-23995) for publication in PLOS ONE.

We have carefully reviewed your comments and we are respectfully requesting a review of your decision to reject the manuscript. We believe the points you have raised about the clarity and omissions in the design of this systematic review protocol, written English not being of an academic standard, and the value of the systematic review topic being questionable are not accurate and are comprehensively addressed within the submitted manuscript. We address each point in turn for your clarity.

1. Clarity and omissions in the design of the systematic review protocol

There were several comments indicating the systematic review protocol methodology was unclear, with suggestions for alternative section headings to improve clarity and comprehensiveness. The headings we have used within the submitted systematic review protocol manuscript align exactly with the required headings in the gold standard for reporting of systematic review protocols, the Preferred Reporting Items for Systematic Reviews and Meta-Analyses Protocols (PRISMA-P) statement.

There were also suggestions to change the structure of eligibility criteria to enhance clarity. The structure of eligibility criteria we have used aligns with the Cochrane Prognosis Methods Group, which has established best-evidence guidelines for developing protocols of systematic reviews of prognostic studies such as this.

There were also several comments indicating that some elements of design were omitted. For example, the comments indicate the date of searching each database is missing. However, the search dates of all electronic databases are clearly detailed in the manuscript, from inception to June 30, 2023 in MEDLINE, EMBASE, CINAHL, and Scopus. We believe all required systematic review protocol methodological design components are clearly and comprehensively addressed, which is supported by the completed PRISMA-P checklist submitted with the manuscript.

2. Written English is not an academic standard

Considerable care was taken to ensure that the manuscript adheres to established academic language conventions. We have, with due diligence, followed recognized guidelines for scientific writing, including proper grammar, syntax, accurate terminology, and writing conventions in the PLOS ONE author guidelines. Prior to submission, the manuscript was reviewed numerous times by the supervisory team which consists of three English as a first language speakers. We are confident we can fully address any specific feedback to enhance the clarity of writing.

3. Value of this topic

The value of the topic of this systematic review protocol is questioned. The introduction to the protocol provides a strong rationale for why this systematic review is required and highlights the value of this topic. For example, knowledge of prognostic factors of outcome for low back pain offers a proactive and personalized approach to patient care. Physical measures are an emerging area of research that offers promise for a comprehensive assessment of low back pain to enable improved patient outcomes. Synthesizing the literature on physical measure prognostic factors to predict outcomes in low back pain will provide essential knowledge to aid identification of high-risk patients for targeted allocation of health care resources and support for improved patient outcomes.

The justification for the importance, need and value of this topic may have been missed as the comments indicate the introduction was not read. Further, the comments indicate the results, discussion and conclusions are questionable. However, as this is a protocol for a systematic review, there are no results or conclusions at this stage of research. We respectfully suggest that this misunderstanding may have contributed to the interpretation of questionable value of our manuscript.

We are confident that this systematic review protocol is a high-value topic that will be of interest to PLOS ONE readers.

In summary

Based on our evaluation of the 3 reasons highlighted above for rejection of this manuscript, we kindly request a reconsideration of the reject decision on this systematic review protocol. We would like to emphasize that our manuscript underwent rigorous internal review and revision to ensure that it adheres to the high standards of PLOS ONE in advance of submission. Our team remains committed to delivering quality research that contributes to the academic community and advances the field. We feel strongly this manuscript will be a valuable publication for PLOS ONE given its methodological rigor and value.

If necessary, we are more than willing to work closely with the reviewers and the editorial team to further clarify any aspects of the manuscript that may have been misunderstood or misinterpreted. Our aim is to ensure that our research is accurately assessed based on its scientific merit and contribution.

Thank you for your time and consideration. We eagerly await your response and hope for a positive outcome.

Sincerely,

Rameeza Rashed, on behalf of all authors

---

## [Decision Letter · Decision Letter 1]

31 Oct 2023

PONE-D-23-23995R1Physical measures of physical functioning as prognostic factors to predict outcomes in low back pain: protocol for a systematic reviewPLOS ONE

Dear Dr. Rashed,

Thank you for submitting your manuscript to PLOS ONE. After careful consideration, we feel that it has merit but does not fully meet PLOS ONE’s publication criteria as it currently stands. Therefore, we invite you to submit a revised version of the manuscript that addresses the points raised during the review process.

We look forward to receiving your revised manuscript.

Kind regards,

Shabnam ShahAli, Ph.D.

Academic Editor

PLOS ONE

Journal Requirements:

2. Please identify your study as 'protocol for systematic review and meta-analysis' in the title of your manuscript.

"Unfunded study"

"No authors have competing interests"

6. We notice that your manuscript file was uploaded on Jul 28 2023. Please can you upload the latest version of your revised manuscript as the main article file, ensuring that does not contain any tracked changes or highlighting. This will be used in the production process if your manuscript is accepted. Please follow this link for more information: http://blogs.PLOS.org/everyone/2011/05/10/how-to-submit-your-revised-manuscript/

Additional Editor Comments (if provided):

Reviewers' comments:

Reviewer's Responses to Questions

**Comments to the Author**

1. Does the manuscript provide a valid rationale for the proposed study, with clearly identified and justified research questions?

Reviewer #1: Yes

Reviewer #2: Yes

2. Is the protocol technically sound and planned in a manner that will lead to a meaningful outcome and allow testing the stated hypotheses?

Reviewer #1: Yes

Reviewer #2: Yes

3. Is the methodology feasible and described in sufficient detail to allow the work to be replicable?

Reviewer #1: Yes

Reviewer #2: Yes

4. Have the authors described where all data underlying the findings will be made available when the study is complete?

Reviewer #1: Yes

Reviewer #2: Yes

5. Is the manuscript presented in an intelligible fashion and written in standard English?

Reviewer #1: Yes

Reviewer #2: Yes

6. Review Comments to the Author

You may also provide optional suggestions and comments to authors that they might find helpful in planning their study.

Reviewer #1: Dear Authors

The manuscript is a protocol study of the Physical measures of physical functioning as prognostic factors to predict outcomes in low back pain: protocol for a systematic review. This topic is important in several ways; importance of low back pain, and importance of using physical measures in the diagnosis. The authors describe a robust, well-designed study. That said, the manuscript in its present form is accepted without any general comments for consideration. I have just some specific comments.

- The presentation of the manuscript is reasonable from an English perspective.

1- Write the PRISMA-P version in method section- Design.

2- Write the registration time in PROSPERO.

3- You could use a table to describe the criteria for the study in details.

4- You should use PRISMA flow diagram as Figure 1 in the main manuscript.

In my opinion the article could be accept and publish.

Reviewer #2: Please write a protocol for a systematic review in the title.

Does this protocol include old people with Low back pain. There is no limited age of participants.

7. PLOS authors have the option to publish the peer review history of their article (what does this mean?). If published, this will include your full peer review and any attached files.

Reviewer #1: **Yes: **Mehrnaz Kajbafvala

Reviewer #2: **Yes: **Yes, sure

---

## [Author Response · Author response to Decision Letter 1]

21 Nov 2023

Date: 21-Nov-2023

Subject: Response to academic editor and Reviewers

Dear Academic Editor and Reviewers

Thank you for your review and consideration of our manuscript titled "Physical measures of physical functioning as prognostic factors to predict outcomes in low back pain: protocol for a systematic review" (PONE-D-23-23995) for publication in PLOS ONE. We have carefully reviewed your comments and incorporated the required changes as below.

Academic Editor (Additional requirements)

1. Please ensure that your manuscript meets PLOS ONE's style requirements

Response: We have reviewed and made sure our manuscript meets PLOS ONE’s style requirements, including file naming.

2. Please identify your study as 'protocol for systematic review and meta-analysis' in the title of your manuscript

Response: Physical measures of physical functioning as prognostic factors to predict outcomes in low back pain: protocol for a systematic review.

3. Financial Disclosure

(d) If you did not receive any funding for this study, please state: “The authors received no specific funding for this work.

Response: In the cover letter, “Unfunded study” has been changed to “The authors received no specific funding for this Work”.

4. Competing Interests

Response: In the cover letter, “No authors have competing interests” has been changed to “The authors have declared that no competing interests exist”.

5. Data availability statement 

Response: This manuscript is a protocol therefore data availability doesn't apply.

6. Upload the latest version of the revised manuscript and review of reference list that is complete and correct

Response: Uploaded and the reference list has been checked.

Reviewers Comments

Reviewer 1

1. Write the PRISMA-P version in the method section- Design.

Response: PRISMA-P 2015 has been added to the design.

2. Write the registration time in PROSPERO.

Response: The date of registration (12 April 2023) has been added.

3. You could use a table to describe the criteria for the study in detail

Response: Thank you for your suggestion to use a table to describe the criteria for the study. After careful consideration, we found that table format did not significantly enhance the clarity of the criteria. However, we have focused on improving the formatting, which we believe has effectively enhanced the overall clarity of the study criteria. We appreciate your feedback and hope that these formatting adjustments meet the desired improvement in readability.

4. You should use PRISMA flow diagram as Figure 1 in the main manuscript

Response: The PRISMA flow diagram has been added as a Figure in the manuscript and uploaded as an individual file.

Reviewer 2

1. Please write a protocol for a systematic review in the title

Response: Title includes “Protocol for a Systematic Review”.

2. Does this protocol include old people with Low back pain. There is no limited age of participants

Response: By including participants of all age groups above 18 years, our study ensures the broad applicability of findings, enhancing the relevance and impact of utilizing physical measures of functioning to predict outcomes in low back pain across diverse populations.

---

## [Editor Report · Decision Letter 2]

29 Nov 2023

Physical measures of physical functioning as prognostic factors to predict outcomes in low back pain: protocol for a systematic review

PONE-D-23-23995R2

Dear Dr. Rashed,

We’re pleased to inform you that your manuscript has been judged scientifically suitable for publication and will be formally accepted for publication once it meets all outstanding technical requirements.

Kind regards,

Shabnam ShahAli, Ph.D.

Academic Editor

PLOS ONE
---

## [Editor Report · Acceptance letter]

1 Dec 2023

PONE-D-23-23995R2 

Physical measures of physical functioning as prognostic factors to predict outcomes in low back pain: protocol for a systematic review 

Dear Dr. Rashed:

I'm pleased to inform you that your manuscript has been deemed suitable for publication in PLOS ONE. Congratulations! Your manuscript is now with our production department. 

Kind regards, 

on behalf of

Dr. Shabnam ShahAli 

Academic Editor

PLOS ONE